# FROM TOKENS TO LATTICES: EMERGENT LATTICE STRUCTURES IN LANGUAGE MODELS

**Bo Xiong**
Stanford University, United States
University of Stuttgart, Germany
xiongbo@stanford.edu

**Steffen Staab**
University of Stuttgart, Germany
University of Southampton, United Kingdom
steffen.staab@ki.uni-stuttgart.de

## ABSTRACT

Pretrained masked language models (MLMs) have demonstrated an impressive capability to comprehend and encode conceptual knowledge, revealing a lattice structure among concepts. This raises a critical question: *how does this conceptualization emerge from MLM pretraining?* In this paper, we explore this problem from the perspective of Formal Concept Analysis (FCA), a mathematical framework that derives concept lattices from the observations of object-attribute relationships. We show that the MLM's objective implicitly learns a *formal context* that describes objects, attributes, and their dependencies, which enables the reconstruction of a concept lattice through FCA. We propose a novel framework for concept lattice construction from pretrained MLMs and investigate the origin of the inductive biases of MLMs in lattice structure learning. Our framework differs from previous work because it does not rely on human-defined concepts and allows for discovering "latent" concepts that extend beyond human definitions. We create three datasets for evaluation, and the empirical results verify our hypothesis.

## 1 INTRODUCTION

Masked language models (MLMs) (Devlin et al., 2019) are pretrained to predict a masked token given its bidirectional context. This self-supervised pretraining enables MLMs to encode vast amounts of human and linguistic knowledge (AlKhamissi et al., 2022; Petroni et al., 2019), making them valuable and implicit knowledge bases for a variety of knowledge-intensive tasks (Van Aken et al., 2019). A key feature of MLMs is their ability to capture conceptual knowledge (Wu et al., 2023; Peng et al., 2022; Lin & Ng, 2022), including understanding concepts and their hierarchical relationships, which are essential aspects of human cognition. Despite substantial evidence (Wu et al., 2023; Peng et al., 2022; Aspillaga et al., 2021; Dalvi et al., 2022) supporting MLMs' ability to conceptualize, the emergence of this capability from their pretraining objectives remains unclear.

To interpret this phenomenon, one pitfall of work in this field is the tendency to focus exclusively on human-defined concepts and investigate how MLMs identify and learn these predefined concepts. This is typically achieved by either classifying (Wu et al., 2023; Peng et al., 2022) or clustering (Aspillaga et al., 2021; Dalvi et al., 2022) terms and then mapping them to established human-defined ontologies, such as WordNet (Miller, 1995). However, this approach often overlooks "latent" concepts that extend beyond human definitions. As a result, while these approaches may illuminate the *what*—the discovery of human-defined concepts—they do not adequately explain the *how*—the underlying mechanisms by which conceptualization emerges from the MLMs' pretraining.

In this work, we adopt a mathematical formalization of concepts inspired by philosophical frameworks (Ganter et al., 2005). In this definition, each concept is defined as the abstraction of a collection of objects that share some common attributes. For example, *bird* is defined as a set of objects that *can fly*, *have feather*, *lay eggs*, and so on, while *eagle* is defined as a *subset* of these objects that share a *superset* of these attributes. By characterizing concepts as sets of objects and attributes, the partial order relations between concepts are naturally induced from the inclusion relations of the corresponding sets of objects and attributes, which is independent of human-defined structures.

To investigate the question–*how does the conceptualization emerge from MLM pretraining?* One might argue that MLMs are trained on explicit definitions of concepts (e.g., through handbooks),

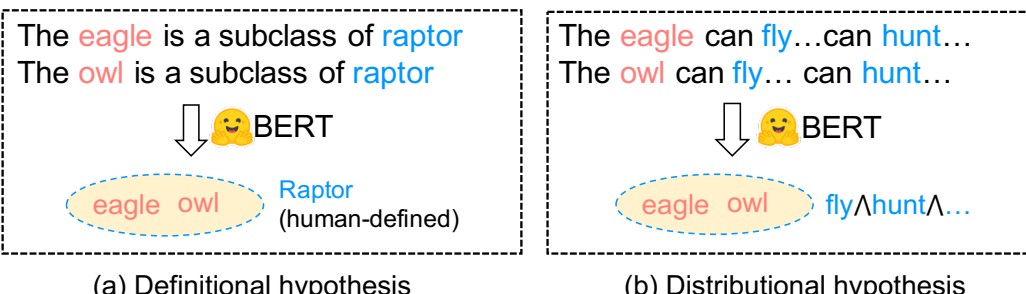

(a) Definitional hypothesis       (b) Distributional hypothesis

Figure 1: **A comparison of two hypotheses of conceptualization in language models.** (a) *Definitional hypothesis* assumes that concepts are learned directly from the definitions (i.e., concepts are explicitly defined in the texts); (b) *Distributional hypothesis* assumes that concepts are learned from the observations of their attributes (i.e., similar concepts have similar attributes).

which we call as the *definitional hypothesis*. While this hypothesis may seem possible, as argued by (Cimiano et al., 2005), such grounding or conceptualization is typically not frequently found in non-wikipedia and non-handbook texts or conversation. Moreover, as shown in Fig. 1(a), the *definitional hypothesis* relies on human-defined concept names as the text "definition", and cannot handle "latent" concepts whose names are not explicitly defined by human. For example, there is no popularly used term specifying birds that can *swim* and *hunt*, but they are indeed a *formal concept* in the context of FCA. Another interpretation is based on the *distributional hypothesis*, which assumes that terms are similar to the extent that they share similar linguistic contexts (Cimiano et al., 2005). Therefore, the *distributional hypothesis* of formal concepts can be interpreted as: concepts are similar to the extent that they share similar contexts describing their attributes. Fig. 1 shows a comparison of the two hypotheses of concept learning for *eagle* and *owl*. Intuitively, the human-defined concept *raptor* is a rarely used term in texts, while the attributes *fly* and *hunt* are more frequently used in texts.

In this paper, we follow the *distributional hypothesis* and argue that *the conceptualization of MLMs emerges from the learned dependencies between the latent variables representing the objects and attributes*. **We made the following contributions:** 1) We show that MLMs are implicitly doing Formal Concept Analysis (FCA), a mathematical theory of lattices for deriving a concept lattice from the observations of object and attributes. This is interpreted as: MLMs implicitly model objects or attributes as masked tokens, and model the *formal context* by the conditional probability between the objects and attributes under certain patterns. As a result, a concept lattice can be approximately recovered from the conditional probability distributions of tokens given some patterns through FCA; 2) we propose a novel and efficient formal context construction method from MLMs using probing and FCA; and 3) we provide theoretical analyses and empirical results on our new datasets that support our findings. Notably, the reconstructed concept lattice contains "latent" concepts that are not defined by human and the evaluation does not require any human-defined concepts and ontologies. Our code and datasets will be made available upon acceptance.

## 2   PRELIMINARIES

**Definition 1** (masked language model). *Let $w := [w_1, \cdots, w_T]$ denote an input sequence of $T$ tokens, each taking a value from a vocabulary $V$, and $w_{\backslash t} := [w_1, \cdots, w_{t-1}, [\text{MASK}], w_{t+1}, \cdots, w_T]$ denote the masked sequence after replacing the $t$-th token $w_t$ with a special $[\text{MASK}]$ token. A masked language model predicts $w_t$ using its bidirectional context $w_{\backslash t}$. Let $\mathcal{D}$ denote the data generating distribution. MLM learns a probabilistic model $p_\theta$ that minimizes the pseudo log-likelihood loss*

$$\text{PLL}_{MLM} = \mathbb{E}_{w \sim \mathcal{D}, t \sim [1, \cdots, T]} - \log p_\theta(w_t | w_{\backslash t}). \tag{1}$$

MLMs can be interpreted as non-linear version of Gaussian graphical models (Wang & Cho, 2019). The idea is to view tokens as random variables that form a fully-connected undirected graph, i.e., by assuming that each variable is dependent on all the other variables, and learn the dependencies across tokens. The dependencies cross tokens can be identified by conditional mutual information (CMI) (Anandkumar et al., 2011), defined as the expected pointwise mutual information (PMI) conditioned

on the rest of the tokens

$$\text{CMI}_{p_\theta}(w_i; w_j|w_{\setminus i,j}) = \mathbb{E}_{w_i,w_j}[\log p_\theta\left(w_i|w_{\setminus i}\left(j, w_j\right)\right) - \log \mathbb{E}_{w_j|w_i} p_\theta\left(w_i|w_{\setminus i}\left(j, w_j\right)\right)], \quad (2)$$

where $w(i, w_j)$ denotes the sentence substituting $w_i$ with a new token $w_j$.

**Formal Concept Analysis** (Ganter et al., 2003) is a principled framework for deriving a concept lattice from a collection of objects and their attributes. FCA follows the mathematical formalization of concepts where each concept is defined as a collection of objects that share some common attributes (Stock, 2010). FCA aims to reconstruct a concept lattice from such observations, which are the objects and their corresponding attributes, represented by a formal context, defined as:

**Definition 2** (formal context). *A formal context is a triple $(G, M, I)$ where $G$ describes a finite set of objects, $M$ describes a finite set of attributes, and $I \subseteq G \times M$ denotes a binary relation (called* incidence*) between objects and attributes with each element $\langle g, m \rangle \in I$ indicating whether an object $g \in G$ possesses a particular attribute $m \in M$.*

A formal context can be represented by a matrix with rows and columns being the objects and attributes, respectively, and each element describing whether the object possesses the corresponding attribute. A formal concept can be defined as a set of objects $G_1 \subseteq G$ sharing a common set of attributes $M_1 \subseteq M$, formally defined as:

**Definition 3** (formal concept). *Given $G_1 \subseteq G$ and $M_1 \subseteq M$, let $G_1' := \{m \in M|(g, m) \in I \,\forall g \in G_1\}$ denote all attributes shared by the objects in $G_1$, and dually $M_1' := \{g \in G|(g, m) \in I \,\forall m \in M_1\}$ denote all objects sharing the attributes in B. The pair $(G_1, M_1)$ is a formal concept iff $G_1' = M_1$ and $M_1' = G_1$. $G_1$ and $M_1$ are called the extent and intent of $(G_1, M_1)$, respectively.*

In a nutshell, the pair $(G_1, M_1)$ is a formal concept iff the set of all attributes shared by the objects in $G_1$ is identical with $M_1$ and dually, the set of objects that share the attributes in $M_1$ is identical with $G_1$. The inclusion between objects/attributes induces a partial order relation of formal concepts.

**Definition 4** (partial order relation). *A partial order relation $\leq_C$ between concepts is defined by*

$$(G_1, M_1) \leq_C (G_2, M_2) \Leftrightarrow G_1 \subseteq G_2 \text{ and } M_2 \subseteq M_1. \quad (3)$$

In this sense, top concept is $\top = (G, G')$ and bottom concept is $\bot = (M', M)$. That is, for any concept $(G_1, M_1)$, we have $(M', M) \leq_C (G_1, M_1) \leq_C (G, G')$. FCA can be extended to deal with three-dimensional data where the object-attribute relations depend on certain *patterns*.

**Definition 5** (triadic formal context (Jäschke et al., 2006)). *A triadic formal context is a quadruple $(G, M, B, Y)$, where $G$ is a finite set of objects, $M$ is a finite set of attributes, $B$ is a finite set of conditions (patterns), and $Y \subseteq G \times M \times B$ is a ternary relation between objects, attributes, and conditions.*

An element $(g, m, b) \in Y$ is interpreted as: The object $g \in G$ possesses the attribute $m \in M$ under the condition $b \in B$. Triadic concept is defined as:

**Definition 6.** *A triadic concept of the triadic formal context $(G, M, B, Y)$ is a triple $(A_1, A_2, A_3)$ where $A_1 \subseteq G$, $A_2 \subseteq M$, and $A_3 \subseteq B$, and the following property holds:*

$$(A_1, A_2, A_3) = \left(\{g \in G \mid \forall m \in A_2, \forall b \in A_3, (g, m, b) \in Y\},\right. \quad (4)$$

$$\{m \in M \mid \forall g \in A_1, \forall b \in A_3, (g, m, b) \in Y\}, \quad (5)$$

$$\left.\{b \in B \mid \forall g \in A_1, \forall m \in A_2, (g, m, b) \in Y\}\right), \quad (6)$$

*where $A_1$ is the set of objects shared by all attributes in $A_2$ under all conditions in $A_3$, $A_2$ is the set of attributes shared by all objects in $A_1$ under all conditions in $A_3$, $A_3$ is the set of conditions under which all objects in $A_1$ share all attributes in $A_2$.*

**Concept Lattice** The set of all concepts and the partial order relation given in a formal context lead to a concept lattice that describes inclusion relationship between their sets of objects and attributes.

## 3 THE LATTICE STRUCTURE OF MASKED LANGUAGE MODELS

We argue that MLMs implicitly model concepts by learning a formal context through conditional distributions between objects and attributes in a target domain. First, we introduce how concept lattices are constructed from MLMs using FCA in Section 3.1. Next, we discuss the inductive biases of MLM pretraining as they pertain to the learning of formal contexts in Section 3.2.

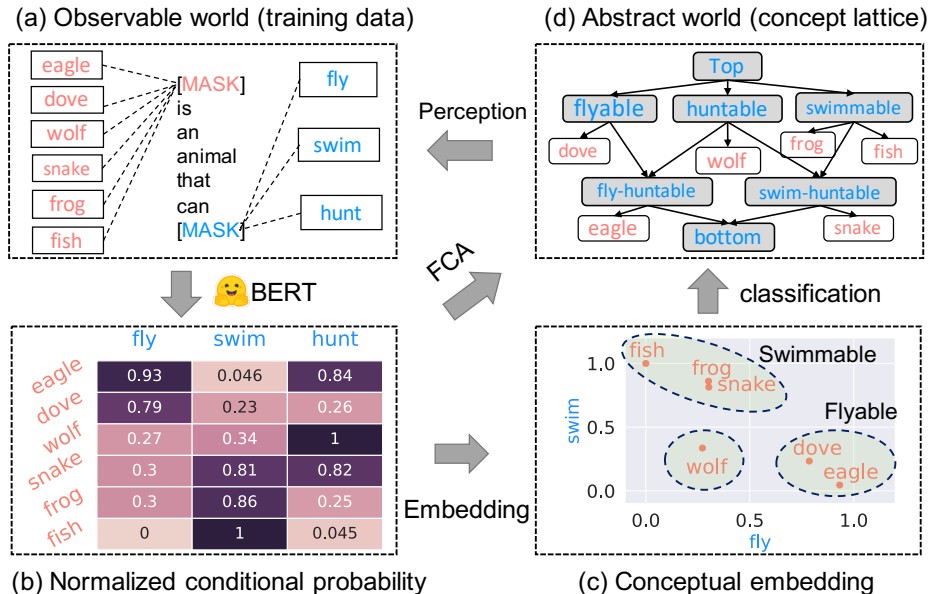

Figure 2: **An illustration of how MLMs learn conceptual/ontological knowledge from natural language**. (a) The observation is a set of sentences that describe relationships between objects and attributes (nouns and verbs in this example). Each sentence is abstracted as a filling of a pattern with an object-attribute pair; (b) the normalized conditional probability between objects and attributes is learned by the MLM (e.g., BERT) and it can be viewed as a formal context in a probabilistic space; (c) each row of the conditional probability can be viewed as the conceptual embedding of the object, and each dimension/column of the embeddings corresponds to a particular attribute. These dimensionally interpretable embeddings can be used for concept classification; (d) the abstract world (concept lattice) can be recovered from the learned formal context in (c) through FCA, and the concept lattice can be viewed as a hierarchy for concept classification.

## 3.1 LATTICE CONSTRUCTION FROM MASKED LANGUAGE MODELS

The relationship between objects and attributes can be explored by *probing* using a cloze prompt, for example, *"[MASK]$_g$ is an animal that can [MASK]$_m$"*, where [MASK]$_g$ and [MASK]$_m$ serve as placeholders for objects and attributes, respectively. This type of template, where both object and attribute tokens are masked, is referred to as a concept *pattern*, denoted by $b$. The instantiation of a pattern with a specific object-attribute pair $(g, m)$ is termed the filling of $b$, denoted as $b^{g,m}$. For partial fillings of the pattern, we use $b^{g,\cdot}$ to denote filling with the object $g$ alone, and $b^{\cdot,m}$ for the attribute $m$ alone. Figure 2a illustrates a concept pattern along with its potential fillings.

**Probabilistic triadic formal context** We estimate the conditional dependencies between objects and attributes using $p_\theta(g|b^{\cdot,m})$ or $p_\theta(m|b^{g,\cdot})$. By cataloguing all possible fillings of objects and attributes within a domain, we construct a conditional probability matrix that serves as an approximation of the formal context for lattice construction. The accuracy of this formal context largely depends on the pattern's efficacy in capturing the intended concept domain and its ability to mitigate confounding influences from other factors. To address these challenges, we utilize multiple patterns and propose the creation of a *probabilistic triadic formal context*, represented as a three-dimensional tensor.

**Definition 7** (construction of probabilistic triadic formal context). *Given a pretrained MLM denoted as $p_\theta$, and considering a set of objects $G$ and a set of attributes $M$ that the model has been trained on (i.e., $G, M \subseteq V$ where $V$ is the vocabulary), and a set of patterns $B$, the triadic formal context $Y$ can be constructed by setting either $Y_{g,m,b} = p_\theta(g|b^{\cdot,m})$ or $Y_{g,m,b} = p_\theta(m|b^{g,\cdot})$, where $g \in G$, $m \in M$, and $b \in B$.*

Fig. 2b provides a normalized representation of a formal context incidence matrix using a single pattern. However, constructing the complete three-way tensor, which requires calculating the conditional probability among any combination of objects, attributes, and patterns, involves a time complexity of $O(|G||M||B|)$. To address efficiency, we propose a method that requires only $|B|\lceil\frac{|G|}{\beta}\rceil$ or $|B|\lceil\frac{|M|}{\beta}\rceil$ evaluations of the cloze prompt in the MLM, with $\beta$ representing the batch size and $\lceil\cdot\rceil$ the ceiling

function. Although the number of possible natural language patterns is infinity in theory, considering the top most informative patterns is sufficient for construction (see Lemma 2). Therefore, it is reasonable to assume that $|G| \gg |M| \gg |B|$, rendering the time complexity nearly linear with respect to the number of objects or attributes. The efficient construction is formalized as follows:

**Definition 8** (efficient construction). *Let $p_\theta(\cdot|w_{\backslash t}) \in \mathbb{R}^{|V|}$ denote the predicted conditional probability vector given $w_{\backslash t}$, and given a set of objects $G$, a set of attributes $M$, and a pattern $b$, we construct $W^{|M| \times T}$ (resp. $W^{|G| \times T}$) such that its $i$th row $W_i = b^{\cdot, m_i}$ (resp. $W_i = b^{g_i, \cdot}$) denote the partial filling of pattern $b$ with attribute $m_i$ (resp. object $g_i$). Let $\mathrm{ind}_G$ (resp. $\mathrm{ind}_M$) be the token indexes of objects $G$ (resp. attributes $M$), the formal context of pattern $b$ can be reconstructed by $\hat{Y}_{\cdot,\cdot,b} := p_\theta(\cdot|W^{|M| \times T})[\mathrm{ind}_G]$ (resp. $\hat{Y}_{\cdot,\cdot,b} := p_\theta(\cdot|W^{|G| \times T})[\mathrm{ind}_M]$).*

The core idea of Def. 8 is to first extract the conditional probability over all possible tokens (i.e., $p_\theta(\cdot|W^{|M| \times T})$ or $p_\theta(\cdot|W^{|G| \times T})$) in a batch manner, and then index the relevant conditional probabilities using the attribute indices $\mathrm{ind}_M$ or object indices $\mathrm{ind}_G$. This construction assumes that either objects or attributes are included in the vocabulary, while the counterpart may consist of multi-token text. Alternatively, instead of relying on a pre-defined set of objects and attributes within a domain, one might generate the set of object-attribute pairs in the domain from the distribution. However, direct sampling from the joint distribution via Markov chain Monte Carlo (MCMC) sampling is challenging. By considering that MLMs can function as probabilistic generative models, we employ Gibbs sampling to sample a sequence of object-attribute pairs that can be used to approximate the joint distribution, which does not presuppose a known set of objects and attributes.

**Definition 9** (formal context generation via Gibbs sampling). *Given a MLM $p_\theta$ and a concept pattern $b$, the process begins by sampling an initial object $g_0$ directly from $p_\theta(\cdot|b)$ and an attribute $m_0$ from $p_\theta(\cdot|b^{g_0, \cdot})$. At each subsequent step $t$ (where $t = 1, 2, \ldots$), a new object $g_t$ is sampled from $p_\theta(\cdot|b^{\cdot, m_{t-1}})$ and a new attribute $m_t$ from $p_\theta(\cdot|b^{g_{t-1}, \cdot})$. The conditional probabilities for all objects and attributes are then computed based on the sequence $(g_0, m_0), \cdots, (g_t, m_t)$.*

**Lattice construction**  The reconstructed probabilistic triadic formal context, denoted as $\hat{Y}$, serves as a smoothed version of the discrete triadic formal context. As shown in Fig. 2c, the smoothed formal context can be viewed as a conceptual embedding that encodes the object-attribute relation in a probabilistic space. We consider its 2-dimensional projection, which approximates the probabilistic incidence matrix $\hat{I}$. This approximation is achieved by aggregating over multiple patterns, either through average pooling $\hat{I}_{g,m} = \frac{\sum_{b \in B} \hat{Y}_{g,m,b}}{|B|}$, or max pooling $\hat{I}_{g,m} = \max_{b \in B} \hat{Y}_{g,m,b}$.

As MLM outputs are softmax probabilities that may result in limited positive responses, and FCA requires a binary input, we apply a min-max normalization approach where, given a threshold $\alpha$, the binarization is performed as follows:

$$I_{g,m} = \frac{\log(\hat{I}_{g,m}) - \min\log(\hat{I})}{\max\log(\hat{I}) - \min\log(\hat{I})} > \alpha. \tag{7}$$

After binarization, a concept lattice is constructed by: (1) generating all formal concepts by identifying combinations of objects and attributes that fulfill the closure property defined in Def. 3, and (2) structuring these formal concepts into a lattice based on the partial order relations defined in Def. 4. Fig. 2d illustrates the resulting lattice with $\alpha = 0.5$ and the corresponding objects of each concept. [1]

### 3.2 INDUCTIVE BIASES OF FORMAL CONTEXT LEARNING

We introduce the formal context learning under certain abstractions. Drawing inspiration from natural language concept analysis (Kamphuis & Sarbo, 1998), implying that conceptualization arises from observing the attributes present in objects, we formalize the world model as a ground-truth formal context defined as $\mathcal{F}_0 := (G_0, M_0, I_0)$. As data captures the perception of the world, we formalize each data point as a "sentence" $x$ describing that an object has a particular attribute, which can be viewed a filling of a concept pattern with an object-attribute pair "sampled" from $\mathcal{F}_0$.

---

[1]In FCA, concepts are inherently unnamed. Names, if required, must be determined in a separate post-processing phase.

---

**Algorithm 1** A Simple Formal Context Learning Algorithm

---

    **Input:** A dataset $\mathcal{D}$, a set of objects $G$ and attributes $M$
    **Output:** An estimated formal context incidence matrix $I = [0,1]^{|G| \times |M|}$
  initialize $I = \mathbf{0}^{|G| \times |M|}$
  **for** $w \in \mathcal{D}$ **do**
    **for** $w_i \in \mathbf{w}$ **do**
      **for** $w_j \in w$ **do**
        **if** $w_i \in G, w_j \in M$ **then**
          $I_{\left(G_{w_i}, M_{w_j}\right)} = I_{\left(G_{w_i}, M_{w_j}\right)} + 1$
        **end if**
      **end for**
    **end for**
  **end for**
  normalization: $I = \text{normalize}(I)$
  return $I$

---

**Abstraction 1** (data generation). *Given a formal context $\mathcal{F}_0 := (G_0, M_0, I_0)$ and a set of concept patterns $B$, each data point $x$ is generated by filling a concept pattern sampled i.i.d. from $B$ with an object-attribute pair $(g, m)$ sampled i.i.d. from the formal context $\mathcal{F}_0$. Thus, each data point is represented as $x = b^{g,m}$, where $b \in B$, $g \in G_0$, $m \in M_0$, and $(g, m) \in I_0$.*

Under such an abstraction, formal context learning aims to reconstruct the world model (original formal context) from the data points generated by Abstraction 1. Given that real-world data generation is not as perfect as Abstraction 1 and might be influenced by other factors or noises. We hence acknowledge that an exact reconstruction of the original formal context is improbable. Therefore, we focus on $\epsilon$-approximate formal context learning.

**Definition 10** ($\epsilon$-approximate formal context learning). *Given a set of $N$ data points $\mathcal{D} = \{w^i\}_{i=1}^N$ generated according to Abstraction 1, $\epsilon$-approximate formal context learning seeks to derive a probabilistic formal context incidence matrix $\hat{I}$ from $\mathcal{D}$, such that the distance $d(\hat{I}, I_0) \leq \epsilon$.*

Our hypothesis is that identifying the formal context incidence matrix from the sentences generated from Abstraction 1 is possible. Lemma 1 establishes the feasibility of identifying the formal context from the generated dataset at the population level.

**Lemma 1** (feasibility). *Let $\mathcal{D} = \{w^i\}_1^N$ be a dataset consisting of data points generated by Abstraction 1, then there exists an identification algorithm $F$ mapping a dataset $\mathcal{D}$ to a formal context $I$ such that $F(\mathcal{D})$ converges to the ground-truth formal context $I_0$ as $N \to \infty$ almost surely, i.e.,* Algorithm 1 is an example algorithm. A full proof of Lemma 1 is provided in the appendix, but the core idea is to demonstrate that $I$ converges to $I_0$ almost surely when the number of data points $N \to \infty$ by using the law of large numbers and concentration inequalities.

**MLMs as formal context learner**    The MLM objective can be interpreted as a non-linear version of Gaussian graphical models for latent structure learning (Mohan et al., 2012), which resembles Lasso regression that aims to predict $x_i$ from $x_{\setminus i}$ with each nonzero coefficient corresponding to a dependency weight. Hence, we can formalize MLM objective as formal context learning by:

**Definition 11** (masked formal context learning). *Let $\mathcal{D} = \{w^i\}_1^N$ be a dataset consisting of data points generated by Abstraction 1, the goal of masked formal context learning is to minimize*

$$\text{PLL}_{\text{MFCL}} = -\mathbb{E}_{w \sim D}\left(\log p_\theta(g|b^{\cdot, m}) + \log p_\theta(m|b^{g, \cdot})\right) - \mathbb{E}_{w \sim D, s \in b_{\setminus g, m}}\left(\log p_\theta(s|b^{\cdot \cdot})\right). \quad (8)$$

The first two loss terms directly encourage the capture of the dependency between objects and attributes under certain the pattern $b$, i.e., $p_\theta(g|b^{\cdot, m})$ and $p_\theta(m|b^{g, \cdot})$. The third loss term implies that the quality of the chosen pattern significantly influences the effectiveness of formal context learning. The intuition is that a pattern might convey much irrelevant or confounding information that confuses the learning of object-attribute dependency. We characterize this confounding information as latent variable information $\mathbf{Z}$. Lemma 2 shows the significant role of the patterns and the influence of $\mathbf{Z}$.

**Lemma 2** (role of pattern). *Let $|\text{CMI}_{p_\theta}(g; m|b)|$ denote the conditional MI without latent variables, and $\text{CMI}_p(g; m|\mathbf{Z}; b)$ denote the conditional MI with latent variables $\mathbf{Z}$, we have $|\text{CMI}_{p_\theta}(g; m|b)| - \text{CMI}_p(g; m|\mathbf{Z}; b) \leq 2H(\mathbf{Z}|b)$, where $H(\mathbf{Z}|b)$ is the conditional entropy.*

Table 1: The candidate concept patterns used to construct formal contexts for different datasets.

| Dataset | A set of concept patterns |
|---|---|
| Region-language | $[\text{MASK}]_m$ is the official language of $[\text{MASK}]_g$
The official language of $[\text{MASK}]_g$ is $[\text{MASK}]_m$
$[\text{MASK}]_m$ serves as the official language in $[\text{MASK}]_g$ |
| Animal-behavior | The $[\text{MASK}]_g$ is an animal that can $[\text{MASK}]_m$
The $[\text{MASK}]_g$ is a type of animal that has the ability to $[\text{MASK}]_m$
An animal known as the $[\text{MASK}]_g$ has the ability to $[\text{MASK}]_m$ |
| Disease-symptom | The $[\text{MASK}]_g$ is a disease that has symptom of $[\text{MASK}]_m$
People who infected the disease $[\text{MASK}]_g$ typically has symptom of $[\text{MASK}]_m$
$[\text{MASK}]_m$ is a kind of symptom of the $[\text{MASK}]_g$ |

Lemma 2 implies that the difference between $|\text{CMI}_{p_\theta}(g; m|b)|$ and $\text{CMI}_p(g; m|\mathbf{Z}; b)$ is bounded by the conditional entropy $H(\mathbf{Z}|b)$. This means that if the concept pattern $b$ provides sufficient information that uniquely describes the object-attribute relationship without introducing confounding factors (i.e., $H(\mathbf{Z}|b) \approx 0$ ), the conditional MI directly captures the dependency between objects and attributes. This re-affirms the importance of the quality of patterns and their critical role in accurately capturing the object-attribute relations. A full proof of Lemma 2 is detailed in the appendix.

---

**Takeaway 1.**

The inductive bias of MLMs in lattice structure learning originates from the learned conditional dependency between tokens. The precision of the lattice construction depends mainly on the quality of the chosen pattern in exclusively capturing the object-attribute relationship.

---

## 4 EVALUATION

In this section, we evaluate whether the formal contexts constructed from MLMs align with established gold standards and assess their capability to reconstruct concept lattices. We have developed several gold-standard formal context datasets across various domains for empirical analysis (Sec. 4.1). Specifically, our investigation addresses two key research questions: 1) Can the conditional probabilities in MLMs effectively recover formal contexts (Sec. 4.2)? 2) Can the reconstructed formal contexts be utilized to construct concept lattices (Sec. 4.3)? We substantiate our findings with additional ablation studies and case analyses in Sec. 4.4.

### 4.1 FORMAL CONTEXT DATASETS

We construct three new datasets of formal contexts in different domains, serving as the gold standards for evaluation. Two of them are derived from commonsense knowledge, and the third one is from the biomedical domain. 1) *Region-language* details the official languages used in different administrative regions around the world. This dataset is extracted from Wiki44k (Ho et al., 2018), utilizing the "official language" relation, which is a densely connected part of Wikidata; 2) *Animal-behavior* captures the behaviors (e.g., *live on land*) of animals (e.g., *tiger*). This dataset is constructed through human curation. We compiled a set of the most popular animal names in English, considering only those with a single token. We identified 25 behaviors based on animal attributes that aid in distinguishing them, such as habitat preferences, dietary habits, and methods of locomotion; 3) *Disease-symptom* describes the symptoms associated with various diseases. We extracted diseases represented by a single token and their symptoms from a dataset available on Kaggle[2]. These datasets vary in terms of density and the nature of objects and attributes. In particular, the *Animal-behavior* dataset is notably denser compared to the other two that are relatively sparse. Additionally, these datasets accommodate cases where objects or attributes can consist of multi-token texts. We design three concept patterns for each dataset by varying the positions of objects and attributes, as given in Table 1. The detailed statistics for these datasets are summarized in Table 4 in the appendix.

---

[2]https://www.kaggle.com/datasets/itachi9604/disease-symptom-description-dataset

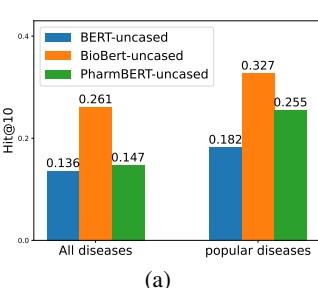
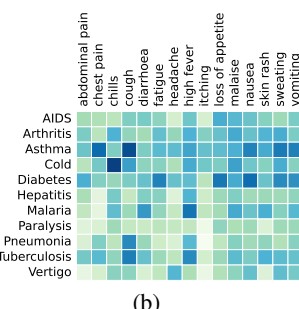
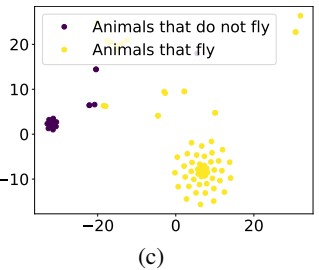

Figure 3: (a) Comparision of BERT, BioBERT, and PharmBERT on formal context learning on disease-symptom; (b) The conditional distribution between diseases and symptoms learned by BioBERT; (c) The T-SNE visualization of the conceptual embeddings constructed from the learned formal context incidence matrix of animals, where the colors denote whether the animals fly or not.

Table 2: The performance of formal context reconstruction on *Region-language* and *Animal-behavior* by the generated conditional probability of three variants of bidirectional MLMs (BERT), where − denotes trivial results whose numbers are $\leq 0.1$. The shaded rows are the baseline results. The best and second best results are **bold** and underline numbers, respectively.

| | | Region-language | | | | Animal-Behavior | | | |
|---|---|---|---|---|---|---|---|---|---|
| | | MRR (↑) | Hit@1 (↑) | Hit@5 (↑) | Hit@10 (↑) | MRR (↑) | Hit@1 (↑) | Hit@5 (↑) | Hit@10 (↑) |
| BERT-base | BertEmb. | 0.340 | 0.249 | 0.403 | 0.478 | - | - | - | - |
| | BertLattice (avg.) | 0.823 | 0.721 | 0.950 | 0.975 | **0.208** | **0.078** | 0.291 | 0.493 |
| | BertLattice (max.) | **0.842** | **0.751** | 0.945 | 0.975 | 0.207 | **0.078** | 0.288 | 0.493 |
| BERT-distill | BertEmb. | 0.094 | 0.002 | 0.109 | 0.214 | - | - | - | - |
| | BertLattice (avg.) | 0.838 | 0.746 | 0.945 | **0.975** | 0.207 | 0.067 | 0.301 | **0.537** |
| | BertLattice (max.) | 0.831 | 0.736 | 0.945 | 0.970 | 0.206 | 0.068 | **0.310** | 0.519 |
| BERT-large | BertEmb. | 0.213 | 0.114 | 0.249 | 0.383 | - | - | - | - |
| | BertLattice (avg.) | 0.833 | 0.731 | **0.970** | **0.975** | 0.194 | 0.067 | 0.276 | 0.453 |
| | BertLattice (max.) | 0.819 | 0.706 | 0.960 | **0.975** | 0.192 | 0.070 | 0.267 | 0.459 |

**Implementation and computational resources** We instantiate our approach (i.e. Def. 8) with BERT and dub it as **BertLattice**. We also design a naive baseline model, dubbed as **BertEmb.**, which predicts object-attribute relations by measuring the distance of the last-layer hidden state embedding vectors of objects and attributes. All experiments were conducted on machines equipped with Nvidia A100 GPU. Our method does not require training of MLMs. The extraction of the formal contexts is very fast ($\leq 60$ seconds) for all three datasets.

## 4.2 CAN CONDITIONAL PROBABILITY IN MLMS RECOVER FORMAL CONTEXT?

We evaluate the capability of MLMs on reconstructing formal contexts from the conditional probabilities by comparing the recovered formal contexts with gold standards. We formalize the problem as a ranking problem and use ranking-based metrics commonly used in ranking problems, specifically mean reciprocal rank (MRR) and hit@k: $\text{MRR} = \frac{1}{N} \sum_{i=1}^{N} \frac{1}{\text{rank}_i}$, hit@k $= \frac{1}{N} \sum_{i=1}^{N} \text{hit}_i$. We consider three variants of BERT models: BERT-distill, BERT-base, and BERT-large. For each model, we use the *uncased* versions and compare the Average pooling and Max pooling variants, denoted as BertLattice (avg.) and BertLattice (max.), respectively.

As Table 2 shows, all variants of BertLattice perform significantly better than BertEmb. on both *Region-language* and *Animal-behavior* datasets. BERT-base performs best in terms of MRR and Hit@1 while BERT-distill and BERT-large outperform BERT-base on *Animal-behavior* and *Region-language* in terms of Hit@5 and Hit@10, respectively. The average results on *Region-language* surpass those of *Animal-behavior*. We hypothesize that this is due to the objects and attributes in *Region-language* being more frequently observed in Wikipedia, where BERT is pretrained. This hypothesis is supported by the observation in Fig. 3(a) showing BERT's relatively lower performance on the domain-specific *Disease-symptom* dataset, while BioBERT and PharmBERT, pretrained on biomedical texts, significantly outperform BERT-base. Additionally, results on popular diseases are better than those on other diseases, suggesting that observation frequency significantly influences MLMs' ability to learn correct object-attribute correspondences. Fig. 3(b) shows part of the *Disease-symptom* formal context extracted from BioBERT. BioBERT accurately identifies symptoms for

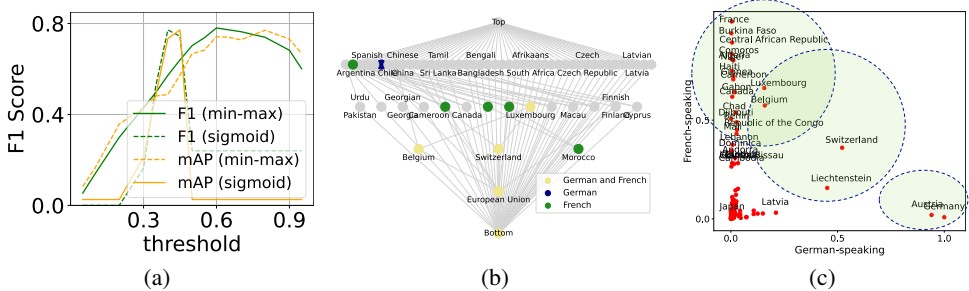

Figure 4: (a) Comparison of different normalization approaches for lattice construction under different thresholds; (b) The reconstructed *Region-language* concept lattice. For visualization, only a part of the concepts and the corresponding objects are labeled. We also highlight German and French speaking lattice paths; (c) The conceptual embeddings of regions/countries in the dimension of German-speaking and French-speaking.

Table 3: The performance (F1 score and mAP) on concept classification, where − denotes trivial results whose numbers are ≤ 0.1. The shaded rows denote the baseline results. The best and second best results are **bold** and underline numbers, respectively.

| | | Region-language | | Animal-behavior | | Disease-symptom | |
|---|---|---|---|---|---|---|---|
| | | F1 (↑) | mAP (↑) | F1 (↑) | mAP (↑) | F1 (↑) | mAP (↑) |
| BERT-base | BertEmb. | 0.214 | 0.027 | 0.269 | 0.393 | - | - |
| | BertLattice (avg.) | 0.630 | 0.528 | 0.643 | 0.388 | - | - |
| | BertLattice (max.) | 0.703 | 0.667 | 0.620 | 0.413 | - | - |
| BERT-distill | BertEmb. | 0.207 | 0.026 | 0.292 | 0.382 | - | - |
| | BertLattice (avg.) | 0.249 | 0.291 | **0.651** | 0.396 | - | - |
| | BertLattice (max.) | 0.703 | 0.667 | 0.617 | **0.417** | - | - |
| BERT-large | BertEmb. | 0.231 | 0.028 | 0.395 | 0.395 | - | - |
| | BertLattice (avg.) | 0.673 | 0.635 | 0.641 | 0.389 | - | - |
| | BertLattice (max.) | **0.709** | **0.701** | 0.636 | 0.407 | - | - |
| BioBERT | BertEmb. | - | - | 0.632 | 0.385 | 0.345 | 0.256 |
| | BertLattice (avg.) | - | - | 0.598 | 0.391 | 0.601 | 0.377 |
| | BertLattice (max.) | - | - | 0.631 | 0.382 | **0.611** | **0.389** |

common diseases, such as *Asthma* with symptoms *Cough* and *chest pain*, and *Cold* with symptom *chills*. Full visualizations of the constructed formal contexts are available in the appendix.

### 4.3 CAN THE RECONSTRUCTED FORMAL CONTEXTS IDENTIFY CONCEPTS CORRECTLY?

We now evaluate whether the formal contexts reconstructed from the conditional probabilities can be used to identify concepts. We frame this as a *multilabel concept classification* task, aiming to identify the correct concepts given an object. Specifically, we classify objects by determining whether they possess all the attributes of a given concept, which is done by measuring the conditional probability under the given patterns. We use F1 score and mean Average Precision (mAP) as metrics, which balance precision and recall and are commonly used in multilabel classification (Xiong et al., 2022).

Table 3 presents the results of concept classification. Generally, we find that all variants of BertLattice outperform BertEmb. on all three datasets. The BERT-base, BERT-distill, and BERT-large models perform well on the *Region-language* and *Animal-behavior* datasets but not as well on the *Disease-symptom* dataset. We hypothesize that this is because these BERT variants are not pretrained on texts involving disease-symptom information. This hypothesis is supported by the observation that BioBERT variants achieve more reasonable results in the *Disease–symptom* data set. An interesting exception is that BioBERT also performs well on the *Animal-behavior* dataset.

### 4.4 ABLATION STUDIES AND VISUALIZATION

**Influence of normalization & threshold**   We investigate the impact of different normalization methods on the performance of lattice construction from the conditional probability. We compare two normalization techniques: min-max normalization, which normalizes at the matrix level, and

sigmoid normalization, which normalizes at the element-wise level. As shown in Fig. 4(a), sigmoid normalization achieves optimal results when the threshold $\alpha \approx 0.5$, whereas min-max normalization performs best when $\alpha \approx 0.6$. Notably, different from the sigmoid normalization, further increasing the threshold of the min-max normalization does not significantly degrade performance, highlighting the robustness and advantages of min-max normalization.

**Visualization**  Fig. 4(b) visualizes the concept lattice constructed from the *Region-language* datasets. The model discovers some latent concepts that humans have not predefined. For example, it identifies the concept of "German and French speaking regions," which includes Switzerland, Luxembourg, Belgium, etc. These are parent classes of the European Union because it includes regions where both German and French, among other languages, are spoken. Importantly, this *Region-language* concept lattice is not predefined by human ontology. Fig. 4(c) and Fig. 3(c) further illustrate the conceptual embeddings derived from the conditional distributions of MLMs. The first figure shows the T-SNE embeddings and demonstrates that the flyable animals and non-flyable animals are clearly distinguished. Fig. 3(c) shows that the "German and French speaking regions" concept corresponds to a set of countries that speak both German and French in the embedding space.

## 5  RELATED WORK

**Conceptual knowledge in language models**  Pretrained MLMs have demonstrated capabilities in capturing conceptual knowledge (Wu et al., 2023; Lin & Ng, 2022). A variety of methods have been proposed to probe MLMs for such conceptual knowledge. Most of them use binary probing classifiers (Aspillaga et al., 2021; Michael et al., 2020) or hierarchical clustering (Sajjad et al., 2022; Hawasly et al., 2024) to identify concepts and validate these against established human-defined ontologies like WordNet (Miller, 1995). Despite these methods offering empirical insights into the conceptual structures encoded by MLMs, they predominantly explore what MLMs learn without addressing how the masked pretraining objectives facilitate the encoding of these conceptual structures. Moreover, it is argued that MLMs can develop novel concepts that do not align strictly with existing human-defined ontologies (Dalvi et al., 2022). This implies that traditional methods may not capture the full extent of conceptual understanding that MLMs are capable of. From a theoretical perspective, recent studies suggest that language models represent concepts through distinct directions in the latent spaces of their hidden activations (Chanin et al., 2024). This observation aligns with the linear representation hypothesis, implying that attributes or features are represented as directional vectors within a model's hidden activations (Elhage et al., 2022; Park et al., 2024). Our work is highly different from previous research, as we consider a formal definition of concepts without any reliance on human-defined concepts and ontologies, and we leverage a mathematical framework (FCA) to analyze concepts.

## 6  CONCLUSION

This paper investigates *how conceptualization emerges from the pretraining of MLMs* through the lens of FCA, a mathematical framework for deriving concept lattices from observations of object-attribute relationships. We demonstrate that MLMs implicitly learn the conditional dependencies between objects and attributes under certain patterns, which can be interpreted as a probabilistic formal context that facilitates the reconstruction of the underlying concept lattice. We then propose a novel framework for lattice construction and discuss the origin of the inductive bias for lattice structure learning. Notably, our analysis of the conceptualization capabilities of MLMs does not rely on predefined human concepts and ontologies, allowing for the discovery of latent concepts inherent in natural language. Our empirical findings on three new datasets support our hypothesis.

**Limitations and future work**  Currently, our FCA framework of MLMs is limited to single-relational data. An immediate future work is to extend FCA to multi-relational data by using multi-relational concept analysis (Wajnberg et al., 2021). For instance, a "Pizza lattice" could be constructed based on both the toppings and sauces of pizzas. Our lattice analysis could also be adapted to autoregressive language models, like GPT series (Ye et al., 2023), which predict the next token given all preceding ones. However, this adaptation requires the masked token for either objects or attributes to be placed at the end of the concept pattern, which may not fit well in some domains.

## REPRODUCIBILITY STATEMENT

Our study does not involve any training or hyperparameter tuning, but instead relies on pre-trained masked language models. We use the PyTorch Transformer library for all model implementations.[3] Our code and datasets are included as supplemental materials and will be made available upon acceptance. Proofs of the lemmas presented in this paper are also provided in the appendix in the supplemental materials.

## ACKNOWLEDGEMENT.

This work was partially funded by the Deutsche Forschungsgemeinschaft (DFG) in the project "COFFEE" — STA 572_15-2.

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

Table 4: Statistics of the used datasets, where density denotes the percentages of positive responses.

|  | #objects | #attributes | density | object token | attribute token |
|---|---|---|---|---|---|
| Region-language | 165 | 45 | 2.59 | single or multi-token | single-token |
| Animal-behavior | 354 | 25 | 40.3 | single-token | multi-token |
| Disease-symptom | 122 | 33 | 6.76 | single-token | single or multi-token |

## APPENDIX

### SUPPLEMENTAL RESULTS

**Computational resources** All experiments were conducted on machines equipped with 4 Nvidia A100 GPU. Our method do not require training of MLMs. The extraction of the formal contexts for our datasets is very fast ($\leq 60$ seconds).

### PROOF OF LEMMAS

**Lemma 1** (feasibility). *Let $\mathcal{D} = \{w^i\}_{i=1}^N$ be a dataset consisting of data points generated by Abstraction 1, then there exists an identification algorithm $F : \mathcal{D} \to I$ such that $F(\mathcal{D})$ converges to the ground-truth formal context $I_0$ as $N \to \infty$ almost surely, i.e., $F(\mathcal{D}) \to I \approx I_0$.*

---

**Algorithm 1** A Formal Context Learning Algorithm

> **Input:** A dataset $\mathcal{D}$, a set of objects $G$ and attributes $M$
> **Output:** An estimated formal context incidence matrix $I = [0,1]^{|G| \times |M|}$
> initialize $I = \mathbf{0}^{|G| \times |M|}$
> **for** $w \in \mathcal{D}$ **do**
>     **for** $w_i \in \mathbf{w}$ **do**
>         **for** $w_j \in w$ **do**
>             **if** $w_i \in G, w_j \in M$ **then**
>                 $I_{\left(G_{w_i}, M_{\mathbf{w}_j}\right)} = I_{\left(G_{w_i}, M_{w_j}\right)} + 1$
>             **end if**
>         **end for**
>     **end for**
> **end for**
> normalization: $I = \text{normalize}(I)$
> return $I$

---

*Proof.* The algorithm 1 is constructed to derive a formal context $I$ from $\mathcal{D}$, Our target is to demonstrate that $I$ converges to $I_0$ almost surely when the number of data points $N \to \infty$. Let $\Omega$ denote the sample space, $I_N$ represent the learned formal context from $\mathcal{D}_N$, and $X_N = d(I_N, I_0)$ denote the random variable indexed by $N$. We need to establish that $X_N \xrightarrow{\text{a.s.}} 0$.

Let us define $E_N := \{\omega \in \Omega : X_N(\omega) > \epsilon\}$ for $\epsilon > 0$, where $\omega$ represents an element of the sample space. Let $Y_{g,m,t} = \begin{cases} 1 & g, m \in x_t \\ 0 & \text{otherwise} \end{cases}$, and

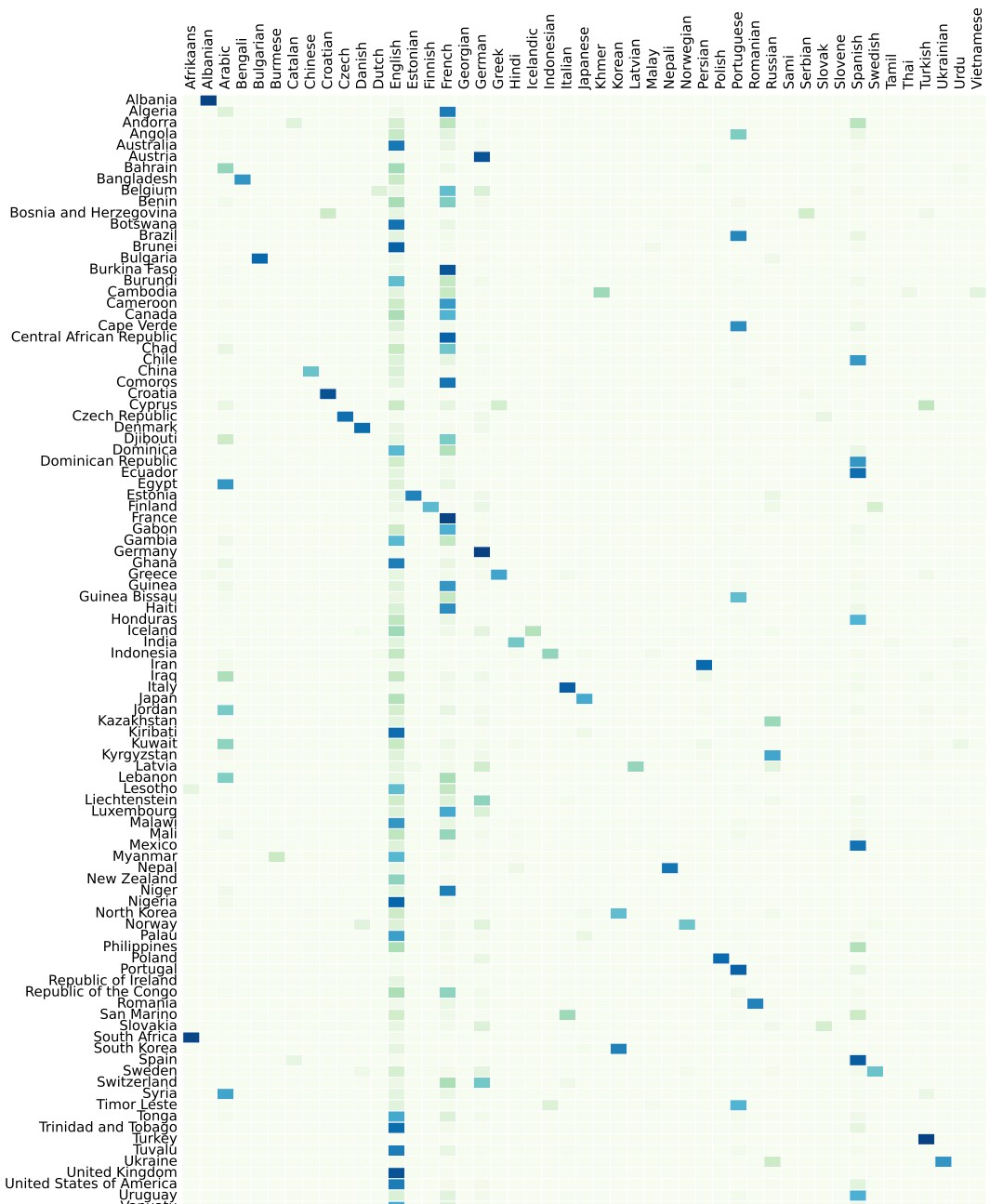

Figure 5: The normalized conditional probability of regions and their official language. The probability is generated by the cloze prompt *"[object] is the official language of [attribute]"*.

consider $\quad P\left(E_N\right) \quad = \quad P\left(\sum_{g\in G, m\in M}\left|\frac{1}{N}\sum_{t=1}^{N}Y_{g,m,t}-I_{0g,m}\right|>\epsilon\right),$ we have

$$P\left(E_N\right) \overset{(a)}{\leq} P\left(\bigcup_{g\in G, m\in M}\left|\frac{1}{N}\sum_{t=1}^{N}Y_{g,m,t}-I_{0g,m}\right|>\frac{\epsilon}{q}\right)$$

$$\overset{(b)}{\leq} \sum_{g\in G, m\in M} P\left(\left|\frac{1}{N}\sum_{t=1}^{N}Y_{g,m,t}-I_{0g,m}\right|>\frac{\epsilon}{q}\right)$$

$$\overset{(c)}{\leq} 2q\exp\left(-\frac{2N\epsilon^2}{q^2}\right),$$

where inequalities (a) and (b) use the union bound, and (c) applies the Hoeffding's inequality.

By applying the Borel-Cantelli lemma, we have $P\left(\limsup_{N\to\infty} E_N\right) = 0$. Hence, $P\left(\lim_{N\to\infty} X_N = 0\right) = 1$. This means that $I$ converges to $I_0$ almost surely when $N\to\infty$. $\qquad\square$

**Lemma 2** (role of pattern). *Let* $|\mathrm{CMI}_{p_\theta}(g; m|b)|$ *denote the conditional MI without latent variables and* $\mathrm{CMI}_p(g; m|\mathbf{Z}; b)$ *denote the conditional MI with latent variables* $\mathbf{Z}$*, we have* $|\mathrm{CMI}_{p_\theta}(g; m|b)| - \mathrm{CMI}_p(g; m|\mathbf{Z}; b) \leq 2H(\mathbf{Z}|b)$*, where* $H(\mathbf{Z}|b)$ *is the conditional entropy.*

*Proof.* Proposition 3 from Zhang & Hashimoto (2021) shows that the dependency between two tokens can be captured by conditional MI. Our lemma can be proved in a similar way by viewing that the objects and attributes are all tokens in the vocabulary. That is $g, m \in V$.

Using the definition of conditional MI, we start with:

$$\mathrm{CMI}_p(g; m|\mathbf{Z}; b) = \mathrm{CMI}_p(g; m|b) - \mathrm{CMI}_p(g; \mathbf{Z}|b) + \mathrm{CMI}_p(g; \mathbf{Z}|m, b)$$

Expanding this, we get:

$$\mathrm{CMI}_p(g; m|\mathbf{Z}; b) = \mathrm{CMI}_p(g; m|b) + H(\mathbf{Z}|g, b) - H(\mathbf{Z}|b) \\ + H(\mathbf{Z}|m, b) - H(\mathbf{Z}|g, m, b).$$

Now, let's consider the difference:

$$|\mathrm{CMI}_{p_\theta}(g; m|b)| - \mathrm{CMI}_p(g; m|\mathbf{Z}; b) \\ = \mathrm{CMI}_p(g; m|b) - \mathrm{CMI}_p(g; m|\mathbf{Z}; b) \\ = -H(\mathbf{Z}|g, b) + H(\mathbf{Z}|b) - H(\mathbf{Z}|m, b) + H(\mathbf{Z}|g, m, b)$$

Next, apply the inequality properties of entropy:

$$\mathrm{CMI}_p(g; m|b) - \mathrm{CMI}_p(g; m|\mathbf{Z}; b) \leq H(\mathbf{Z}|b) + H(\mathbf{Z}|g, m, b)$$

Since entropy is always non-negative, we further have:

$$H(\mathbf{Z}|g, m, b) \leq H(\mathbf{Z}|b)$$

Combining these, we get:

$$|\mathrm{CMI}_{p_\theta}(g; m|b)| - \mathrm{CMI}_p(g; m|\mathbf{Z}; b) \leq H(\mathbf{Z}|b) + H(\mathbf{Z}|g, m, b) \\ \leq 2H(\mathbf{Z}|b)$$

Therefore, we conclude that:

$$|\mathrm{CMI}_{p_\theta}(g; m|b)| - \mathrm{CMI}_p(g; m|\mathbf{Z}; b) \leq 2H(\mathbf{Z}|b).$$

This completes the proof. $\qquad\square$

