# OpenReview forum: "From Tokens to Lattices: Emergent Lattice Structures in Language Models"
_ICLR.cc/2025/Conference — ICLR 2025 Poster_

### Official Review · Reviewer_nF1q · 2024-10-28

**Soundness:** 2
**Presentation:** 2
**Contribution:** 2
**Rating:** 6
**Confidence:** 3

**Summary:**

This paper explores how masked language models (MLMs) like BERT implicitly capture conceptual knowledge and organize it into lattice structures. The authors employ Formal Concept Analysis (FCA) to illustrate that MLMs' training objectives inherently form a "formal context" that can be mapped into a lattice framework. The paper diverges from traditional approaches by not relying on human-defined concepts, instead proposing that MLMs capture "latent" or unspoken concepts beyond human ontologies. The author examines this approach in three datasets: region-language, animal-behavior and disease-symptom, and concludes that MLMs are efficient in modeling the concept hierarchies in probabilistic terms.

**Strengths:**

1. The innovative application of FCA enables the reconstruction of concept lattices without relying on human-defined concepts and ontologies, allowing for the discovery of latent concepts that traditional methods might overlook.

2. The paper includes a variety of visualizations that effectively illustrate key concepts, enhancing clarity and engagement.

3. The proposed method is computationally efficient, making it practical for implementation without requiring additional training or significant resources.

4. The paper offers a thorough empirical analysis across diverse domains, which supports the validity of its claims that MLMs can generate meaningful concept lattices.

**Weaknesses:**

1. The study relies on carefully curated datasets that may not capture the complexity or variability of real-world language use. Expanding the dataset to include a broader range of contexts and ambiguities would enhance the framework’s robustness and applicability.

2. While concept lattices offer a structured hierarchy, they are currently limited to modeling simple object-attribute relationships. Incorporating more nuanced or multi-relational knowledge types would broaden the framework’s applicability.

3. The paper’s presentation could be significantly improved. Simplifying the exposition and emphasizing intuitive explanations would make the work more accessible and engaging.

**Questions:**

1. Can this framework be applied to autoregressive language models (e.g., GPT series) instead of masked language models?

2. How are concept patterns selected for this framework? Are they designed empirically by humans, and to what extent could different patterns impact the results?

3. How might this framework perform on larger, more complex datasets with more ambiguous or polysemous relationships?

---

> ### Author Response · Authors · 2024-11-20
> **Response**
>
> Thank you for your valuable comments. Below, we provide our responses.
>
> **Weakness 1**: The study relies on carefully curated datasets that may not capture the complexity or variability of real-world language use. Expanding the dataset to include a broader range of contexts and ambiguities would enhance the framework’s robustness and applicability.
>
> **Answer**: We agree it is important to choose datasets that capture the complexity and variability of real-world language usage. Actually, we used datasets from different domains (e.g., Disease-symptom is a medical domain dataset) and vary in terms of density, single/multi-token, etc, which reflect the real-world language usage. Our datasets are the largest datasets for formal concept analysis.
>
> **Weakness 2**: While concept lattices offer a structured hierarchy, they are currently limited to modelling simple object-attribute relationships. Incorporating more nuanced or multi-relational knowledge types would broaden the framework’s applicability.
>
> **Answer**: Our framework can be easily extended to multi-relational data. We just have to design suitable patterns for different relations and extract the relationships with corresponding patterns. However, constructing lattices from a multi-relational formal context is time-consuming, as the time complexity is exponential due to the combinatorial nature of the subset search across multiple dimensions.
>
> **Weakness 3**: Emphasizing intuitive explanations would make the work more accessible and engaging.
>
> We understand that the writing of our paper is more mathematically rigorous than other papers that also work on language model probing. This is because our contribution includes also theoretical analysis. But we agree some parts of it can be improved with some intuitive explanations. To echo this, we will revise the descriptions of the method to include more intuitive explanations.
>
>
>
>
> **Question 1**: Can this framework be applied to autoregressive language models (e.g., GPT series)?
>
> **Answer**: Yes. To do so, we only have to put either the attribute or object tokens at the end of the context. This is because autoregressive models can only predict the next-token probability.
>
> **Question 2**: How are concept patterns selected for this framework? Are they designed empirically by humans, and to what extent could different patterns impact the results?
>
> **Answer**: Yes, they are designed by humans, but in a way such that the patterns reflect how the objects and attributes are used.
>
> **Question 3**: How might this framework perform on larger, more complex datasets with more ambiguous or polysemous relationships?
>
> **Answer**:  According to our Lemma 2, the performance depends on how well the patterns capture the real-world usage of the objects and attributes, but does not depend on the size of the datasets. If there are more ambiguous or polysemous relationships between objects and attributes or if the pattern does not capture enough information of the relationships, then the probability of their corresponding relationships will also become more uncertain, e.g., 0.5.

---

> > ### Comment · Reviewer_nF1q · 2024-11-25
> >
> > Thank you for the authors' response. This addresses most of my concerns, and I have raised my score accordingly. I look forward to seeing more intuitive explanations included in the final version.

---

> ### Author Response · Authors · 2024-11-27
> **Thank you for raising your score**
>
> Thank you very much for raising your score. We will definitely include more intuitive explanations in our final version.

---

### Official Review · Reviewer_D1N8 · 2024-10-30

**Soundness:** 3
**Presentation:** 3
**Contribution:** 3
**Rating:** 8
**Confidence:** 2

**Summary:**

This paper explores how masked language models (MLMs) like BERT inherently learn conceptual relationships by organising knowledge into "concept lattices" through a framework called Formal Concept Analysis (FCA). The authors demonstrate that through this framework MLMs can reconstruct concept relations directly from language patterns in an more effective and explainable way.

**Strengths:**

- This work introduces a well-structured FCA-based probing framework that translates MLM logits to formal contexts, allowing for insightful analysis.
- The constructed concept lattices effectively recover and represent conceptual knowledge that MLMs encode.
- The framework also ensures efficient construction of concept lattices.

**Weaknesses:**

- The baseline models only consider naive BERT embeddings, omitting a range of established probing techniques known for their effectiveness, such as prompt-based probing in natural language inference (NLI) formats. Incorporating these methods would provide a stronger basis for claiming that this FCA-based framework is more effective in extracting knowledge from MLMs.

- The evaluation is limited to "single-token" named entities, which may restrict dataset coverage. While the `[MASK]` token only predicts one token at a time, several methods exist to approximate multi-token predictions.

**Questions:**

- Have you considered the issue of potential ill-formed entity or relation names fitting into the template, which might introduce bias into evaluation?

- A minor point concerns the definition of MLM. In practice, not every token in a sequence is masked, so it would be more accurate to define MLMs with respect to a set of sampled masked tokens.

---

> ### Author Response · Authors · 2024-11-20
> **Response**
>
> Thank you for your valuable comments. Below, we provide our responses.
>
> **Weakness 1**: The baseline models only consider naive BERT embeddings, omitting a range of established probing techniques known for their effectiveness, such as prompt-based probing in natural language inference (NLI) formats. Incorporating these methods would provide a stronger basis for claiming that this FCA-based framework is more effective in extracting knowledge from MLMs.
>
> **Answer**:  We have added two more baselines: one naive baseline with a template generated by the definitional hypothesis denoted as BertLattice (definitional), and another baseline with natural language inference (NLI) formats denoted as BertLattice (NLI) . The results indeed show that the patterns are very important to the results, and the patterns should reflect how the objects and attributes are used in real-world.
>
> The corresponding templates and results are seen in the following tables:
>
> |  | Region-language | Animal-behavior |
> |---|---|---|
> | BertLattice (definitional) | [Region] is an object that has the official language [language] | [anima] is the object that can [behavior] |
> | BertLattice (NLI)  | Is [language] the official language of [Region], the answer is [Yes/No] | Can [anima] [behavior], the answer is [Yes/No] |
>
> |  | Region-language |  | Animal-behavior |  |
> |---|---|---|---|---|
> |  | F1 | mAP | F1 | mAP |
> | BertEmb | 0.214  | 0.027 | 0.269 | 0.393 |
> | BertLattice (definitional) | 0.372 | 0.211 | 0.276 | 0.375 |
> | BertLattice (NLI) | 0.578 | 0.501 | 0.588 | 0.381 |
> | BertLattice-avg (distribiutional) | 0.630   | 0.528 | **0.643** | 0.388 |
> | BertLattice-max (distribiutional) | **0.703** | **0.667** | 0.620 | **0.413** |
>
> **Weakness 2**: The evaluation is limited to "single-token" named entities, which may restrict dataset coverage. While the [MASK] token only predicts one token at a time, several methods exist to approximate multi-token predictions.
>
> **Answer**: We want to clarify our datasets do contain multi-token objects or attributes. We only limit that either attributes or objects are single-token. For exampl, in the animal-behavior datasets, most of behaviors are multi-token. We also agree that it is possible to approximate multi-token predictions, but this goes beyond our target of this paper and we leave it as our future work.
>
> **Question 1**:  Have you considered the issue of potential ill-formed entity or relation names fitting into the template, which might introduce bias into evaluation?
>
> **Answer**: According to our Lemma 2, the performance depends on how well the patterns capture the relationships between the objects and attributes. So if there are ill-formed entity or relation names that are ambiguous or polysemous, it is also hard to find a good template/pattern that uniquely captures the ill-formed entity or relation names, so the uncertainty of their corresponding relationships will also be high. This is due to the statistical nature of the language model training.

---

> > ### Comment · Reviewer_D1N8 · 2024-11-21
> >
> > I appreciate the authors' efforts in addressing my concerns, particularly with the additional experiments exploring different NL patterns. However, I would appreciate further justification for favouring the FCA framework over standard probing methods, such as directly using MLM logits.
> >
> > While I am generally inclined toward accepting the paper, I acknowledge that my expertise in FCA is limited, and as such, I will maintain a low confidence score.

---

> > > ### Author Response · Authors · 2024-11-27
> > > **Further justification**
> > >
> > > Thanks for your response. We favor FCA based method because FCA does not rely on human-defined ontology and concepts and can discover any latent concepts, while standard probing methods rely on human-defined ontology for evaluation. This is really important as our goal is to study the mechanism of MLMs and our findings suggest that MLMs learn concepts in a way that contradicts how humans learn concepts. So we should not use human-defined ontologies to evaluate our method.
> > >
> > > We hope these have addressed your concerns and we appreciate if you can raise your rating and/or confidence.

---

> > > > ### Comment · Reviewer_D1N8 · 2024-11-27
> > > >
> > > > Thank you for providing further clarification. I encourage you to emphasise this aspect more in the revised version of the paper. I have increased my review score.

---

> > > > > ### Author Response · Authors · 2024-11-29
> > > > > **Thank you for increasing your scores**
> > > > >
> > > > > Thank you very much for increasing the scores. We will definitely emphasize this aspect in our final version.

---

### Official Review · Reviewer_t7YG · 2024-11-04

**Soundness:** 2
**Presentation:** 2
**Contribution:** 2
**Rating:** 6
**Confidence:** 3

**Summary:**

In the paper “From Tokens to Lattices: Emergent Lattice Structures in Language Models” submitted to ICLR 2025, the authors answer the question whether Formal Concept Analysis  concept lattices can be extracted in a meaningful way from an MLM, such as BERT. They answer this question positively, which is demonstrated by a dedicated experiment.

**Strengths:**

I find the paper interesting to read. It studies a sound problem about an ability of LLMs to perform Symbolic AI tasks.

**Weaknesses:**

Definition 5 is vogue, it is more like an intuition rather than definition

In L (line) 263, it is said that data generation (i.e., Abstraction 1) is an inherently noisy process. However, I do not see anything noisy in Abstraction 1, everything is nice and clean. What is meant by noisy here?

In L 253, Animal-behaviour dataset is presented. But where does the data come from? Is it just the authors’ beliefs and nothing more?

In L 374, it is said “We formalize the problem as a recommendation system problem”. But this is not clear at all without further details, and so the essence of the first experiment is not clear for me.

In the introduction and the conclusion, it is emphasised that the suggested method does not depend on predefined human concepts and ontologies. However, it is misleading: it clearly depends on humans, who have to provide patterns B.


Minor:
L 53: extra whitespace after “possible”
L 99: D should be \mathcal D and p_\theta should be defined
L 125: Do not start a sentence with a math symbol, especially if the previous one ends with such a symbol
L 211: Start the definition with “Given a pattern b”
L 237: b \in B should be in the subscript
L 243: g, m should be in the subscript
L 268: it should be defined what is the distance in this context
L 275, 278: bf w should be just w
L 285: hypothesis
L 290: standard notation F: D \to I says that F is a function with domain D and range I. This is clearly not what you want to say here. Also, D should be \mathcal D.

**Questions:**

Every comment in the weakness section, except the list of minor corrections, is essentially a question.

---

> ### Author Response · Authors · 2024-11-20
> **Response**
>
> Thank you for your comments. Below, we provide our responses.
>
> **Weakness 1**: Definition 5 is vogue, it is more like an intuition rather than definition
>
> **Answer**: In our revision, we have split the definition 5 (triadic formal context and triadic formal concept) into two separate definitions and make them more clear and formal.
>
> **Weakness 2**: In L (line) 263, it is said that data generation (i.e., Abstraction 1) is an inherently noisy process. However, I do not see anything noisy in Abstraction 1, everything is nice and clean. What is meant by noisy here?
>
> **Answer**: You are right. Abstraction 1 is indeed a clean and perfect process. What we meant by “noisy” is that real-world data generation is not always as perfect as Abstraction 1 and there are some other noises influencing the generation. Hence, our reconstruction (Def. 9) is only approximately possible. We paraphrased the description to avoid confusion.
>
> **Weakness 3**:  In L 253, Animal-behaviour dataset is presented. But where does the data come from? Is it just the authors’ beliefs and nothing more?
>
> **Answer**: Animal-behaviour was constructed by humans with external knowledge (e.g., wikipedia). We construct it by both humans (for very easy cases such as dogs can fly?) and verify them by checking their Wikipedia description.
>
> **Weakness 4**:  In L 374, it is said “We formalize the problem as a recommendation system problem”. But this is not clear at all without further details, and so the essence of the first experiment is not clear for me.
>
> **Answer**: Recommendation system is a ranking problem, i.e., given a user, ranking all possible items. Our problem is essentially also a ranking problem, i.e., given an object, rank all possible attributes or vice versa. We paraphrase this sentence to make it more clear.
>
> **Weakness 5**:  In the introduction and the conclusion, it is emphasised that the suggested method does not depend on predefined human concepts and ontologies. However, it is misleading: it clearly depends on humans, who have to provide patterns B.
>
> **Answer**: What we meant is that our method can discover concepts that are not defined by humans and the evaluation does not depend on human ontologies. Our algorithm does depend on a human-designed pattern, but a pattern is just a part of the algorithm so it is just like a hyperparameter users give to the algorithm.
>
> **Minor issues**: We fixed all minor issues in the writing. Thanks for the very detailed suggestions.

---

> ### Author Response · Authors · 2024-11-29
> **Any further comments?**
>
> Dear reviewer t7YG,
>
> Thank you again for your thoughtful review and constructive feedback.  We wanted to kindly remind you to review our rebuttal and share any further thoughts or updates. We provided straightforward answers to your questions and have addressed your comments about the writings in our revised version.
>
> If these have addressed your concerns, we kindly ask you to increase your score. Also, please let us know if there is any additional clarification or further information we can provide. This is very important to us as we are eager to ensure that all points have been adequately addressed to your satisfaction.
>
> Best regards
>
> Authors

---

### Official Review · Reviewer_yYtU · 2024-11-09

**Soundness:** 2
**Presentation:** 2
**Contribution:** 2
**Rating:** 6
**Confidence:** 4

**Summary:**

The authors propose a framework that uses Formal Concept Analysis (FCA), to explore the conceptual understanding of an MLM (Masked Language Model) and their hierarchical relationships.
FCA is a well-known mathematical framework researched in the early 2000's or so. FCA aims to uncover concept lattices from attribute-value data.
MLM are trained neural networks that aim to predict a masked token in a sentence given its context. They are known to be capable of contextual understanding that can be used as pre-training in various NLP tasks.
The big question the authors pose is: how does conceptualization emerge from pre-training an MLM?
The authors hypothesize that MLMs learn the conditional dependencies between objects and attributes under certain patterns. Then they propose a framework to probe MLMs through the use of FCA to construct a lattice of latent concepts without relying on human-defined ontologies. Their empirical evaluation, using three datasets they developed, showed that conditional probabilities in MLM can help recover formal contexts and construct concept lattices.

**Strengths:**

+ Bridging the gap between FCA and MLM is a good idea that shows that old and new AI can work in tandem, especially when it comes to the interpretability of black-box models and their usefulness in subsequent tasks. Using FCA could be a nice step in this direction.

+ Paper well-written in general, appropriate citations.

+ The formalization seems appropriate.

+ The ability to uncover latent concepts without human ontologies gives a chance to non predefined concepts to emerge and hence has the potential to increase our knowledge in the domain.

**Weaknesses:**

- There is a lack of general motivation on why it is important to uncover the lattice of the concepts embedded in an MLM and how it will be used. If the goal is XAI, the authors should highlight it.

-  The scalability of the framework to multi-relation is unclear and using one single relation seems quite limiting. While it is mentioned by the authors as a limitation, a discussion about how limiting the lattice to one relation can miss patterns would be good. The authors could discuss the challenges that might arise when tackling multi-relations. This might include the complexity of the lattice itself.

- The experimental section is limited in the number and choice of the datasets.

- Figures are hard to read.

**Questions:**

1-  How can the lattice of concepts be leveraged in NLP tasks?
2-  Does using only one relation mean that FCA does not allow capturing the full extent of the conceptual understanding the MLM is capable of?
3- How are the uncovered concepts labeled?

---

> ### Author Response · Authors · 2024-11-20
> **Response**
>
> Thank you for your valuable comments. Below, we provide our responses.
>
> **Weakness 1**: There is a lack of general motivation on why it is important to uncover the lattice of the concepts embedded in an MLM and how it will be used. If the goal is XAI, the authors should highlight it.
>
> **Answer**: Our goal is indeed XAI. In particular, we aim to understand the mechanisms of language models in learning conceptual knowledge. We have highlighted this point in our revision.
>
> **Weakness 2**: The scalability of the framework to multi-relation is unclear and using one single relation seems quite limiting. While it is mentioned by the authors as a limitation, a discussion about how limiting the lattice to one relation can miss patterns would be good. The authors could discuss the challenges that might arise when tackling multi-relations. This might include the complexity of the lattice itself.
>
> **Answer**: To handle multi-relational data, we just need to design separate patterns for different relations and apply multi-relational FCA. Everything is done then.  The challenges would be to design good-quality patterns for different relations as they may have confounding factors that influence each other and the increase of the computational complexity . We add this discussion in the conclusion section.
>
> **Weakness 3**: The experimental section is limited in the number and choice of the datasets.
>
> **Answer**:  Unfortunately, there are no existing datasets suitable for evaluating our framework.  We have created three datasets with human curation. They vary in terms of domains, single/multi-tokens, density, etc.
>
> **Question 1**: How can the lattice of concepts be leveraged in NLP tasks?
>
> **Answer**: The lattice of concepts from FCA is a structured semantic hierarchy from unstructured text or corpora. All NLP tasks that require a structured representation of knowledge (e.g., ontologies, taxonomies) such as question answering and named entity recognition, entity typing can benefit from the lattice of concepts.
>
> **Question 2**: Does using only one relation mean that FCA does not allow capturing the full extent of the conceptual understanding the MLM is capable of?
>
> **Answer**: No, FCA can be generalised to multi-relational data, e.g., Pizza can be classified based on both its toppings and sauces. However, such generalization requires more computation and it is less useful in practice. We left this extension as our future work.
>
>
> **Question 2**: How are the uncovered concepts labelled?
>
> **Answer**: In FCA, concepts are unlabelled (that is why do not assume human-defined concepts). Labels, if required, can be named using all the attributes it has. For example, the set of objects that can fly and hunt can be named “swim-huntable”.

---

> ### Comment · Reviewer_yYtU · 2024-11-27
>
> Thanks for addressing my questions. I think your work is promising and therefore, I raised my score.

---

### Official Review · Reviewer_RPj7 · 2024-11-11

**Soundness:** 2
**Presentation:** 2
**Contribution:** 2
**Rating:** 1
**Confidence:** 4

**Summary:**

The work focuses on generating concept lattices from masked language models such as BERT, RoBERTA, etc. The idea is to utilize the information contained in these models for generating a partial ordering of the concepts instead of learning the concepts from textual information.

**Strengths:**

--> Concept lattices are directly constructed from masked language models.

**Weaknesses:**

--> It should be made clear in the paper if the authors are directly constructing the concept lattice or the formal context first and then generating the lattice. This makes a difference since generating concept lattices starting from a formal context is computationally expensive.
--> The contributions/research questions of the paper should be explicitly mentioned in the introduction.
--> Why not using Large Language Models for constructing lattice and comparing it to masked language models since the masked language models are not fine-tuned they are rather used by masking the tokens.
--> The broader applicability of these lattices is a bit unclear, where would these lattices be used? If the objective is to classify the terms into the classes then this can be done without passing through the concept lattice generation phase. i understand that there is a partial ordering between the concepts but there is a concept of hierarchical classification.
--> Introduction should have a clear example/scenario for motivating the objectives of this work.

**Questions:**

--> Are authors directly generating a concept lattice or generating a formal context?
--> What is the computational complexity of generating the context and then the lattice?
--> What is the broader applicability on where these lattices can be used for solving which tasks?
--> Why not prompt Large Language Models which contain much more information?

---

> ### Author Response · Authors · 2024-11-20
> **Response**
>
> Thank you for your comments. It seems that the reviewer misunderstands the significant aspects of the paper. Below, we provide our responses.
>
> **Question 1**: directly generating a concept lattice or generating a formal context
>
> **Answer**: We generate formal context first and then derive the concept lattice with FCA. FCA does require a formal context as input.
>
> **Question 2**: Computational complexity of generating the context and then the lattice?
>
> **Answer**: The time complexity of context generation is linear with respect to the number of objects or attributes (e.g., O(|M|) or O(G) depending on which one is masked to predict). The time complexity of lattice construction, in the worst case, is indeed exponential in the size of the input formal context, as there is as most 2^{min(|G|, |M|)} number of formal concepts. **Although FCA suffers from computation complexity issues, this does not hurt our contribution**. Our contribution is NOT to obtain SoTA results on concept lattice construction, but rather to understand and interpret the mechanism of MLMs (i.e., how do MLMs learn conceptual knowledge). Our major finding is that MLMs learn conceptual knowledge similarly to FCA, and we verify our findings by FCA. Of course, we fully agree that FCA is computationally expensive but that does not contradict our findings.
>
> **Question 3**: The contributions/research questions of the paper should be explicitly mentioned in the introduction.
>
> **Answer**: To echo your suggestion, we have highlighted our contributions/research questions more explicitly in our revision with colors.
>
> **Question 4**: Why not using Large Language Models？why not prompt Large Language Models which contain much more information?
>
> **Answer**: We agree that larger models may obtain better results. However, our goal is NOT to seek SoTA results on lattice construction but rather to understand and interpret the behaviors or mechanisms of masked language models. So at this point, we choose to use BERT because it is the most popular masked language models.
>
>
> **Question 5**: The broader applicability of these lattices is a bit unclear, where would these lattices be used? If the objective is to classify the terms into the classes then this can be done without passing through the concept lattice generation phase.
>
> **Answer**:  First, the lattice of concepts from FCA is a structured semantic hierarchy from unstructured text or corpora. All NLP tasks that require a structured representation of knowledge (e.g., ontologies, taxonomies) such as question answering and named entity recognition, entity typing can benefit from the lattice of concepts.
>
> Second, our objective is NOT to obtain a good lattice, but rather to understand, interpret, and verify the behaviors or mechanisms of masked language models. In particular, we investigate how MLMs learn conceptual knowledge. FCA is just a mathematical tool used to verify our findings.

---

> ### Author Response · Authors · 2024-11-27
> **Rebuttal period is ending**
>
> Dear reviewer RPj7,
>
> The deadline of the author-rebuttal session is approaching. We sincerely appreciate if you can take a look at our response and reconsider your score. We believe our response has effectively addressed your concerns. If you have any other concerns, please do not hesitate to contact us.

---

> > ### Author Response · Authors · 2024-12-02
> > **Kind reminder to respond to our rebuttal**
> >
> > Dear reviewer RPj7,
> >
> > We wanted to remind you that today is the last day that reviewers can reply to authors, we kindly ask you to review our point-to-point response and our changes in the revised submission.
> >
> > **It will only take you minutes to read and respond, as our answers to you questions/concerns are actually very straightforward and concise.**
> >
> > Best

---

### Meta-Review · Area_Chair_Qhpi · 2024-12-23

**Metareview:**

This paper uses Formal Concept Analysis (FCA) to derive concept lattices and explore hierarchical relationships in masked language models (MLMs). A majority of reviewers are in favor of acceptance, and there is a general consensus that the ideas presented here are interesting and relevant for XAI in MLMs. I recommend acceptance as a poster in the conference.

**Additional Comments On Reviewer Discussion:**

Several reviewers participated in the discussion, with a couple increasing their score.

---

### Decision · Program_Chairs · 2025-01-22

Accept (Poster)